# Effect of Influenza Vaccination on Rate of Influenza Virus Infection in Chinese Military Personnel, 2015–2016: A Cluster Randomized Trial

**DOI:** 10.3390/vaccines11091439

**Published:** 2023-08-31

**Authors:** Yapin Li, Jianxing Yu, Qingfeng Li, Dan Yu, Wenjing Song, Qi Liu, Dongqi Gao, Qiulan Chen, Haiyang Zhang, Liqun Huo, Jian Wang, Jiayi Wang, Huisuo Yang, Gang Zeng

**Affiliations:** 1Central Theater Command Center for Disease Control and Prevention, No. 66 Heishitou Road, Beijing 100042, China; yapin0215@163.com (Y.L.); lqfnds69@sina.com (Q.L.); 18010003393@163.com (W.S.); gaodq1217@163.com (D.G.); lazysheepsea@163.com (H.Z.); wellginger1987@126.com (J.W.); 2Sinovac Biotech Ltd., No. 39, Shangdi West Road, Beijing 100085, China; yujx7939@sinovac.com (J.Y.); yud@sinovac.com (D.Y.); chenql@sinovac.com (Q.C.); huolq@sinovac.com (L.H.); 3Sinovac Life Sciences Ltd., Beijing 102601, China; liuq8245@sinovac.com (Q.L.); wangjy1755@sinovac.com (J.W.)

**Keywords:** trivalent influenza vaccine, cluster randomized trial, surveillance, incidence rate, military personnel, China

## Abstract

Influenza is a major cause of morbidity and mortality. The protective effect of a trivalent influenza vaccine (TIV) is undetermined in military personnel. We conducted an open-label, cluster randomized trial on active-duty servicemen of Beijing, Tianjin, and Shijiazhuang, who were randomly assigned to receive either a single dose of TIV or no treatment, according to cluster randomized sampling. The subjects were then followed for a maximum of six months to assess the incidence of laboratory-confirmed influenza. A total of 5445 subjects in 114 clusters received one dose of TIV before the 2015/2016 influenza season. Laboratory-confirmed influenza was identified in 18 in the vaccine group compared with 87 in the control group (6031 subjects in 114 clusters), resulting in a vaccine effectiveness (VE) of 76.4% (95%CI: 60.7 to 85.8) against laboratory-confirmed influenza. Influenza-like illness was diagnosed in 132 in the vaccine group compared with 420 in the control group, resulting in a VE of 64.1% (95%CI: 56.2 to 70.6). The estimated VE against influenza B viruses was 80.5% (95%CI: 65.6 to 88.9) and 8.6% (95%CI: −241 to 75.5) against influenza A viruses. In conclusion, the trivalent influenza vaccine is moderately effective, highly immunogenic, and generally safe to use in healthy male military servicemen.

## 1. Introduction

Influenza causes considerable morbidity and mortality worldwide, resulting in an estimated 1 billion influenza cases, with 3–5 million severe illnesses and 290,000–650,000 influenza-related respiratory deaths (0.1–0.2% case fatality rate) each year [1,2]. In China, an estimated annual mean of 88,100 influenza-associated excess respiratory deaths were caused by seasonal influenza between 2010 and 2015 [3]. Vaccination is considered the most effective strategy for preventing influenza [4,5,6,7]. Offering seasonal influenza vaccination in national immunization can not only reduce morbidity and mortality but also strengthen capacities for pandemic influenza preparedness and response. The U.S. Advisory Committee on Immunization Practices recommends the routine annual influenza vaccination of all persons aged ≥6 months [8]. The World Health Organization (WHO) recommends annual seasonal influenza immunization, primarily for those at a high risk of developing severe influenza-associated disease and death (i.e., individuals with comorbidities and underlying conditions, older adults, and pregnant women) and those at an increased risk of exposure to or transmission of influenza virus (i.e., health workers), as well as people living in congregate living settings, such as prisons, refugee camps, and group homes [2]. Military personnel are prone to outbreaks of influenza for a variety of reasons, including crowding and stressful conditions [9]. Although not recommended by the WHO, an influenza vaccination program has been recommended to military personnel in the USA since 1946 [10] and is associated with a moderately lower risk of influenza [11]. However, there are considerable data gaps existing in the disease burden and cost reduction of influenza vaccination in this specialized group, especially in middle-income countries. The influenza vaccination of military personnel is not recommended or is only offered on a voluntary basis in most countries [12,13], including China [14].

Anflu^®^, a preservative-free, unadjuvanted, split-virion, trivalent inactivated influenza vaccine (TIV), was introduced by Sinovac Biotech Ltd. for marketing in China in 2006 [15] and approved for use among children and adults aged six months and above based on immune titers observed in one pivotal clinical trial [16]. The vaccine has shown to be highly immunogenic and has confirmed a satisfactory safety profile in several post-marketing studies [17,18,19,20], including one conducted among military personnel [20]. However, since the regulatory authority in the country can accept hemagglutination inhibition (HI) titers as a surrogate of protection to facilitate the annual approval of a seasonal influenza vaccine, considerable knowledge gaps exist regarding the protective effectiveness of TIVs against clinically apparent influenza, despite their wide distribution and administration in domestic populations and abroad [15]. In addition, no data is available on the effect of TIVs against influenza among military personnel, which is ideally necessary for optimizing vaccine use or for recommending immunization programs in such specific groups living in highly restricted and confined settings.

In an attempt to provide post-licensure effectiveness data with TIVs administered systematically to a population of healthy young adults, as well as to inform military vaccination policies in the country, this open-label, cluster randomized, phase Ⅳ clinical trial compared the incidence of laboratory-confirmed influenza among active-duty serviceman between those who received one dose of TIV and those who received no treatment before the start of the 2015–2016 influenza season.

## 2. Materials and Methods

### 2.1. Study Site

The study was conducted in three cities of North China (i.e., Beijing, Tianjin, and Shijiazhuang) between 1 October 2015 and 30 April 2016. Before the study began, we approached the three sites and gained permission of participation from twelve brigades (seven in Beijing, two in Tianjin, and three in Shijiazhuang), including 233 companies and 15,525 active-duty servicemen.

All the active-duty servicemen in the sites were offered the opportunity to participate based on the following inclusion/exclusion criteria. Healthy individuals aged 18 years or older were eligible for inclusion if they were not participating in clinical trials of other drugs or vaccines during the study period and if they did not use immunosuppressive agents (e.g., corticosteroids) in the month before or during the trial. The key exclusion criteria were individuals with acute exacerbation of chronic diseases (including chronic hepatitis, hypertension, diabetes, heart disease, etc.); who had active malignant tumors or tumor recurrence after treatment; who were immunocompromised (e.g., HIV/AIDS); who had a reported fever or an axillary temperature of more than 37.0 °C; who had a history of Guillain–Barre Syndrome; who had a vaccination history of influenza vaccines of any type in the 2015/2016 flu season; who were pregnant or lactating; and who had a known allergy to any vaccine component.

### 2.2. Primary and Secondary Outcomes

The primary endpoint was laboratory-confirmed influenza in the vaccinated subjects and nonrecipients of study vaccine occurring during the follow-up period (from 21 days to six months after vaccination or corresponding visit for controls). The secondary endpoints included influenza-like illness (ILI) and severe influenza occurring during the follow-up period. Other safety and immunogenicity endpoints were also evaluated, including adverse reactions occurring within 0–3 days and 0–21 days after vaccination in the vaccine group and hemagglutination inhibition (HI) antibody responses to four influenza subtypes measured at day 21 after vaccination in the vaccinated subjects and nonrecipients of the study vaccine.

We defined ILI as a reported fever or a measured axillary temperature of at least 38 °C, with cough or sore throat, according to one national surveillance case definition by the Chinese Center for Disease Control and Prevention [21]. To specifically measure the incidence of influenza, an ILI case testing positive for influenza virus using real-time reverse transcriptase polymerase chain reaction (RT-PCR) was considered laboratory-confirmed influenza. Severe influenza was defined as influenza with at least one severe symptom or syndrome (as determined according to Ministry of Health-issued guidelines [22]; see Appendix A).

We calculated the seroconversion rate, geometric mean titers (GMTs), geometric mean increase (GMI), and seroprotection rate for HI antibodies. Seroconversion was defined based on baseline antibody titer of subjects (day 0), i.e., post-immune antibody titer (day 21 for control) ≥ 1:40 if baseline antibody titer < 1:10 or post-immune antibody titer increased ≥ 4 folds if baseline antibody titer ≥ 1:10. The GMI was calculated as the ratio of post-immune antibody titer and baseline antibody titer. The cut-off threshold for seroprotection of HI antibodies was set at ≥1:40.

### 2.3. Vaccines

Anflu^®^ was derived from influenza viruses recommended by the World Health Organization for the northern hemisphere during the 2015–2016 season [23], i.e., A/Switzerland/9715293/2013 (H3N2)-like virus, B/Phuket/3073/2013 (B/Yamagata lineage)-like virus, and A/California/7/2009 (H1N1) pdm09-like virus, cultured in chicken embryo, followed by harvest, inactivation, purification, and disruption. The vaccines were provided in 0.5 mL prefilled single-dose syringes, containing 15 μg of hemagglutinin (HA) for each strain (lot number: 201505011).

### 2.4. Procedures

#### 2.4.1. Randomization of Clusters

One company (mean cluster size = 70, range = 15 to 230) served as the unit of randomization in the study. A statistician independent of the study assigned the 223 companies (i.e., clusters) at random with an allocation ratio of 1:1 to either the vaccine group or no-treatment control group, using a standard computer pseudorandom number generator. The study was open-labeled, and no blinding was used. In addition, two clusters (one in vaccine group and one in control group) were randomly selected among the companies of Beijing as the immunogenicity subgroup to evaluate HI antibody responses.

#### 2.4.2. Administration of Study Vaccine

Before the influenza epidemic starting in North China [24,25], i.e., between 1 October and 20 October, 2015, all participants in the vaccine group who met the eligibility criteria for enrollment and gave informed consent were administered one 0.5 mL dose vaccine by subcutaneous injection at the attachment of the lateral deltoid muscle of the upper arm. Vaccination was administrated by trained and qualified nurses at the sites’ military health care center.

#### 2.4.3. Follow-Up of Adverse Events

Participants were observed in the study site for at least 30 min immediately after vaccination. Paper diary cards were given to vaccine recipients to help record any injection-site adverse events (e.g., pain, erythema, swelling, itching, and induration) or systemic adverse events (e.g., headache, fatigue, nausea, vomiting, diarrhea, and myalgia) occurring within 0–3 days of vaccination. Predefined symptoms (solicited events) and other unspecified symptoms (unsolicited events) reported by the participants in the vaccine group were recorded. On the day 3 visit, diary cards were collected by investigators. From day 4 to day 21 after vaccination, safety data were collected from spontaneous reports from the participants combined with the regular visit. Any serious adverse events were reported up to 6 months after vaccination for vaccine recipients only. All adverse events were assessed by study investigators for severity according to the China National Medical Products Administration guidelines [26]. The causality between adverse events and vaccination was determined by the investigators. Any adverse event categorized as possibly, probably, or definitely related to vaccination was defined as an adverse reaction.

#### 2.4.4. Active Surveillance for Influenza-like Illness (ILI) Cases

Active surveillance of ILI was established in the military health care center before study began at the three sites. The occurrence of ILI among the vaccinated subjects and non-recipients of study vaccine were actively reported via the established ILI surveillance system over the follow-up period in the sites. In the country, active-duty military personnel must seek care via the military health care system, which allows us to capture near all ILI cases in the sites with this system. Doctors in the military health care system were trained to diagnose and report ILI using a paper form. For each ILI case identified in the system, throat or nasal swabs were collected immediately after the medical encounters and were sent to the central laboratory at Central Theater Command Center for Disease Control and Prevention to test for influenza viruses within 48 h of collection. All the laboratory-confirmed influenza cases were followed to date of recovery or death.

#### 2.4.5. Laboratory Methods

At the central laboratory, viral RNA was extracted from the throat or nasal swabs and tested for the presence of influenza virus subtypes (i.e., influenza virus type A, influenza virus type B, influenza A of subtype H1 and subtype H3), using RT-PCR assays according to the National Technical Guidelines for Surveillance of Influenza issued by Chinese National Influenza Center (2011 version) [27].

To assess the HI antibody responses, 5 mL of whole blood without adding any anticoagulants was collected from the immunogenicity subgroups on day 0 (immediately before vaccination) and day 21 after vaccination. Blood samples were centrifuged to separate serum, which was prepared into three aliquots and stored at −20 °C temperature until the time of analysis. Hemagglutination inhibition (HI) assay was used to determine antibody titers to influenza subtypes as reported previously [28]. Briefly, serum samples were treated with receptor-destroying enzymes (RDE, cholera filtrate, Sigma-Aldrich, Saint Louis, MO, USA) at 37 °C for 16–18 h, bathed at 56 °C for 50 min to inactivate any remaining RDE, cooled down to room temperature, mixed with 50 µL of packed turkey red blood cells (RBCs), and incubated overnight in a 2–8 °C refrigerator before titration. Then, the supernatant of RDE-treated serum was tested in 2-fold dilution starting with a 1:5 dilution. A standardized quantity of HA antigen (4 units) was mixed with serially diluted serum and incubated at room temperature for 45 min. Then, 1% turkey RBCs were added, and the degree of binding of the antibody to the HA molecule was assessed 30 min later. The titers were expressed as the reciprocal of the highest dilution that showed the complete inhibition of hemagglutination, and HI titers below 1:5 were assigned an arbitrary value of 1:2.5. All the control antigens and reference anti-sera used in the assay were purchased from the National Institute for Biological Standards and Control (NIBSC), London, UK.

### 2.5. Study Oversight

Participants provided written informed consent before enrolment. The clinical trial protocol and informed consent form were approved by the Ethics Committee of Central Theater Command Center for Disease Control and Prevention (approval no., Center [2015]-102). The study was conducted in accordance with the requirements of Good Clinical Practice of China, the Declaration of Helsinki, and local regulatory requirements [29,30] and reported in compliance with the Consolidated Standards of Reporting Trials (CONSORT) 2010 statement.

### 2.6. Statistical Analysis

We assumed that the risk of influenza in the control group during our study period would be approximately fifteen cases per 1000 subjects (calculated as an overall 3% attack rate divided by 2) [31], the intra-cluster correlation coefficient would be 0.01, and the mean and the coefficient of variation of the cluster size would be 70 and 0.832, respectively. Allowing for a dropout rate of 10%, we calculated that two groups, each with approximately 7492 subjects and 108 clusters, would be sufficient to detect a risk reduction of 50% with a power of 80% at a two-sided significance level of 0.05.

We conducted all analyses in SAS (version 9.0). For the comparison of individual-level and cluster-level variables at baseline, we used Student’s *t*-test or the Wilcoxon test for continuous variables and the chi-square test or Fisher’s exact test for categorical variables as appropriate. For the analyses of vaccine protection, the effectiveness of vaccination (VE) was estimated as ([1 − relative risk] × 100%). The reactogenicity analysis was conducted for participants who received one dose of the vaccine. Adverse events were summarized descriptively as frequencies and percentages by time interval, type of event, and severity for the vaccine group only. The immunogenicity analysis was performed on the immunogenicity subgroup, who had both HI antibody results at baseline and the values at day 21 after vaccination. We calculated 95% CIs for all categorical outcomes using the Clopper–Pearson method and for HI antibody GMTs based on the t-distribution of the log-transformed antibody titer.

## 3. Results

### 3.1. Characteristics of Participants

In total, 233 trial clusters were approached and agreed to participate in the trial (Figure 1). After randomization, 118 clusters with 7838 subjects were allocated to the vaccine group, and the rest (115 clusters with 7687 subjects) were allocated to the control group. Within the vaccine group, 5445 were vaccinated and 2393 were unvaccinated as their data at baseline were not available because of refusal to participate or sign an informed consent form; within the control group, 1567 were excluded for the same reasons. During the follow-up period, five companies were disbanded and were lost to follow-up (four in vaccine group and one in control group). Finally, 5279 vaccinated subjects in the vaccine group and 6031 nonrecipients in the control group were included in the final analysis, resulting in a dropout rate of 24% for the whole study cohort (33% in the vaccine group and 22% in the control group).

Characteristics of participants and clusters were similar in the two groups (Table 1). The median age of study subjects was 21 years (range: 18–38 years), and 99% of participants were male. The majority of study participants had a high school education level (62%). At the cluster level, the median number of persons in the cluster was 61 (interquartile range, IQR = 39–89). In the vaccine group, the median number of persons vaccinated in the cluster was 36 (IQR = 20–65), and the vaccination coverage was 70% (IQR = 52–89%).

### 3.2. Total Protection

During the follow-up period, 552 episodes of ILI (incidence risk 4.88%, 95% CI = 4.49 to 5.29) occurred (Figure 2). The ILI occurrence was significantly different between the three sites; i.e., Beijing had 220 episodes (incidence risk 4.16%, 95% CI = 3.64 to 4.73), Tianjin had 252 episodes (incidence risk 7.46%, 95% CI = 6.59 to 8.40), and Shijiazhuang had 80 episodes (incidence risk 3.03%, 95% CI = 2.41 to 3.75; *p* < 0.001). ILI was identified in 132 participants in the vaccine group (incidence risk 2.50%, 95% CI = 2.10 to 2.96) compared with 420 participants in the control group (incidence risk 6.96%, 95% CI = 6.33 to 7.64), resulting in a vaccine protective effectiveness against ILI for the influenza vaccine of 64.1% (95% CI: 56.2 to 70.6) (Table 2).

Respiratory specimens of all the 552 ILI cases were tested for, and influenza viruses were identified in 105 (19%) participants (96 with influenza B virus, 3 with influenza A of subtype H1, 2 with influenza A of subtype H3, and 4 with influenza A of un-typable) (Figure 2). Laboratory-confirmed influenza were identified in 18 participants in the vaccine group compared to 87 in the control group, resulting in a vaccine protective effectiveness against influenza for the influenza vaccine of 76.4% (95% CI: 60.7 to 85.8) (Table 2). Protection against types of influenza virus circulating in the 2015/16 season was also studied. During the study’s follow-up period, influenza B viruses were detected in 96 participants with 14 in the vaccine group and 82 in the control group, resulting in a significant vaccine protective effectiveness against the influenza B virus for the influenza vaccine of 80.5% (95% CI: 65.6 to 88.9). In contrast, the influenza A virus was identified in only 9 participants in the trial (4 in vaccine group and 5 in control group). For protection against the influenza A virus circulating in the season, the effectiveness of the influenza vaccine was estimated at 8.6% (95% CI: −241 to 75.5) (Table 2).

No deaths or severe influenza cases were identified. Compared with the control group, participants who had received one dose of the TIV reported a less frequent headache (5.6% vs. 37.9%, *p* = 0.006) and myalgia (0% vs. 40.2%, *p* = 0.001) when they had laboratory-confirmed influenza during the follow-up periods (Appendix A).

### 3.3. Serologic Responses

In total, the immunogenicity subgroup included 149 subjects (67 in vaccine group and 82 in control group). At baseline, participants in the two study groups had similar geometric mean titers and seroprotection rates of serum HI antibodies against the three influenza virus strains included in the vaccine formulation (A/H1N1, A/H3N2, and B/Yamagata) (Figure 3 and Appendix A). After vaccination on day 21, the HI antibody responses against the three influenza virus strains were significantly increased in the vaccine group. On day 21 after vaccination, the seroconversion rates of HI antibodies against three influenza virus strains (i.e., A/H1N1, A/H3N2, and B/Yamagata) were 68.7%, 85.1%, and 86.6%, respectively; the seroprotection rates were 98.5%, 100%, and 97.0%, respectively; and the GMIs were 7.7, 8.4, and 11.4, respectively (Figure 3 and Appendix A). The HI antibodies against the B/Victoria strain (not included in the formulation) increased only slightly.

### 3.4. Safety and Reactogenicity

The vaccine was generally well tolerated among healthy adults in the trial. Overall, 25 (0.46%) subjects experienced a reaction among the 5445 subjects who received one dose of the investigational vaccine, including swelling at the injection site and fever (21, 0.39%), rash (1, 0.02%), Henoch–Schonlein purpura (1, 0.02%), psychogenic reaction (1, 0.02%), and syncope (1, 0.02%). All the reactions were classified by investigators as mild in severity and resolved by the end of the study.

## 4. Discussion

We conducted a large-scale, cluster randomized trial with a TIV among active-duty military personnel over one influenza season (2015–2016) in North China. To our knowledge, this is the first study that investigated the protective effectiveness of a TIV against influenza-related morbidity among military servicemen in the country. The TIV could provide 76.4% protection against laboratory-confirmed influenza and 64.1% protection against ILI in the vaccinated subjects. The vaccine also showed satisfactory immunogenicity and safety profiles compared with other studies conducted in previous influenza seasons [15]. Together, our study provided robust and convincing evidence that the TIV can be effectively and safely used in male military servicemen.

During our study, the TIV demonstrated strong prevention against influenza-related morbidities in our study participants. A recent Cochrane meta-analysis review of 52 clinical trials concluded that well-matched inactivated vaccines provide moderate protection in healthy adults, with a 59% (95% CI 51% to 66%) effectiveness for preventing influenza and a 16% (95% CI 5% to 25%) effectiveness for preventing ILI [6]. Among active-duty military servicemen in the USA, a similar result was observed with a moderate effect of inactivated influenza vaccines against influenza-associated medical encounters during three consecutive seasons (28.4–54.8%) [11]. However, the UK’s study on influenza vaccination in its armed forces population revealed a lower protective effect of a 15% reduction in preventing ILI encounters [32]. In our study, the protective effect of the TIV was a little higher compared with those reports. Numerous confounders could exist and modify the vaccine effectiveness between studies, including host factors, variations in study design, and differences between seasons or vaccines, including strain mismatch or egg adaptation [2]. The following could be some of the reasons for our high protection. First, our study was conducted among a population of previously unvaccinated, healthy, young adults, which could result in a higher vaccine effectiveness than in those who had previously received repeated flu vaccinations [11,33]. Another explanation is that our study might have overestimated the vaccine effect as our study subjects had high dropout rates, especially in the vaccine group (33%) compared to the control group (22%). We guess that the high dropout rate in the vaccine group might have an impact on the VE estimates as those who have higher health awareness (and lower risk of contracting influenza) were more likely to be kept in the vaccine group. Finally, the predominance of influenza B viruses (91%, 96/105) circulating among the study subjects might also have had an effect on the vaccine protection rates observed in our study. According to the Chinese National Influenza Center data [34], during the study period between 1 October 2015 and 30 April 2016, influenza B viruses were co-circulating with other strains in North China, accounting for 62% (8474/13725) of all the strains tested for. Less representation of other strains (i.e., influenza A/H1N1 and A/H3N2) in our study subjects and some heterogeneity of the infection risk between study sites suggest that the high level of protection seen in our study should be interpreted with caution. An outbreak caused by influenza B/Yamagata lineage might have been ongoing during our study period in our study population.

Our study demonstrated that the TIV induced high and robust HI antibody titers against the strains contained in the vaccine, i.e., A/H1N1, A/H3N2, and B/Yamagata. These results are highly consistent with other studies conducted in previous seasons [15]. Immunogenicity endpoints have long been used as surrogates of protection by regulators to facilitate the annual approval of seasonal influenza vaccine strain changes [2,35,36]. For a vaccine to be considered sufficiently immunogenic, at least one of the following three criteria has to be met for each of the antigenic strains contained in the vaccine, i.e., a seroconversion rate > 40%; seroprotection rate > 70%; and GMI > 2.5. In our study, the titers of HI antibodies against the three strains contained in the vaccine all reached or were superior to these regulatory criteria. Although the level of HI antibodies against the B/Victoria strain did not meet the immunogenic criteria of protection, it increased slightly 21 days after vaccination, suggesting that the vaccine can also elicit some weak cross-reactivity of antibodies against other strains not included in the vaccine [15]. Unfortunately, our study did not further type the 96 influenza B viruses into linages. We have reason to believe that there had been an outbreak of the B/Yamagata lineage in our study subjects as the vaccine’s effectiveness on influenza B viruses was as high as 80.5% in the study. One previous meta-analysis estimated that the inactivated vaccine provided an average of 63% protection against influenza B virus infection. Interestingly, several studies found that the B/Yamagata lineage has not been isolated in many regions of the world since 2020 April, suggesting that this influenza lineage might have become extinct, which may have far-reaching consequences on the influenza vaccine’s development and reformulation [37]. Continued monitoring on the influenza strain circulation in the national laboratory surveillance program is guaranteed to help discern whether the virus is truly extinct.

A major strength of the study is its large sample size. Over a very short period of three weeks before the influenza epidemic starting, we vaccinated over 5000 subjects in three sites and followed them for six months to prospectively evaluate the effect of vaccination on influenza-related clinical outcomes in the population. In addition, comprehensive laboratory methods, including PCR and serological assays in the central laboratory, which were not available in previous clinical studies, were used to allow us to specifically measure influenza incidences. Finally, our study was conducted among military personnel, a neglected group that has a high risk of influenza outbreaks. Our study could fill in knowledge gaps of this group as less evidence of influenza vaccine protective effectiveness is currently available in this specialized group of people.

Our study has limitations. First, the dropout rate in the study was high and might have potentially undermined the validity of our study results. The major reason for dropout was the refusal of participating or signing an informed consent after randomization since we did not approach individual participants to ask permission to participate. Subsequently, because of dropout after randomization, the indirect protection for influenza provided by the vaccine was not evaluated as it was originally designed in the study. Vaccination of children and adolescents in communities has been shown to offer protection for unimmunized residents against influenza in previous studies [38]. Indirect protection, or so-called herd protection, is a very important attribute of vaccines when planning immunization programs, which can be readily estimated by using a cluster randomized study design [39]. Thus, one lesson learned from our study is that when conducting cluster randomized trials, the willingness of participation of individuals should also be evaluated before study planning to select eligible or qualified sites to conduct the study. Secondly, the study was limited by the strains circulating since our study was conducted in one season, during which influenza B dominated in our study subjects. The effectiveness of the TIV against other strains contained in the vaccine needs to be further evaluated in future studies. The season-to-season variations and duration of protection for the TIV need to be studied in the future. Finally, our open-labeled study design might have introduced some biases. For example, investigators are more likely to report positive results from the vaccine group, and participants in the vaccine group are also more likely to report events, which will bias our results to a certain extent. Nevertheless, this is the first study in which we studied influenza vaccine effectiveness in a real-world setting and among the neglected group of military servicemen, and thus, its reporting to the academic society is warranted.

## 5. Conclusions

Our study showed that Anflu^®^ is moderately effective, highly immunogenic, and generally safe to use in healthy young adults and that it can significantly lower the risk of influenza. Military personnel are prone to outbreaks of influenza. Raising immunity via seasonal vaccination is the key to protect against influenza in this particular group. Studies evaluating the cost-effectiveness of TIVs are warranted for expanding influenza vaccination among military personnel in the future.

## Figures and Tables

**Figure 1 vaccines-11-01439-f001:**
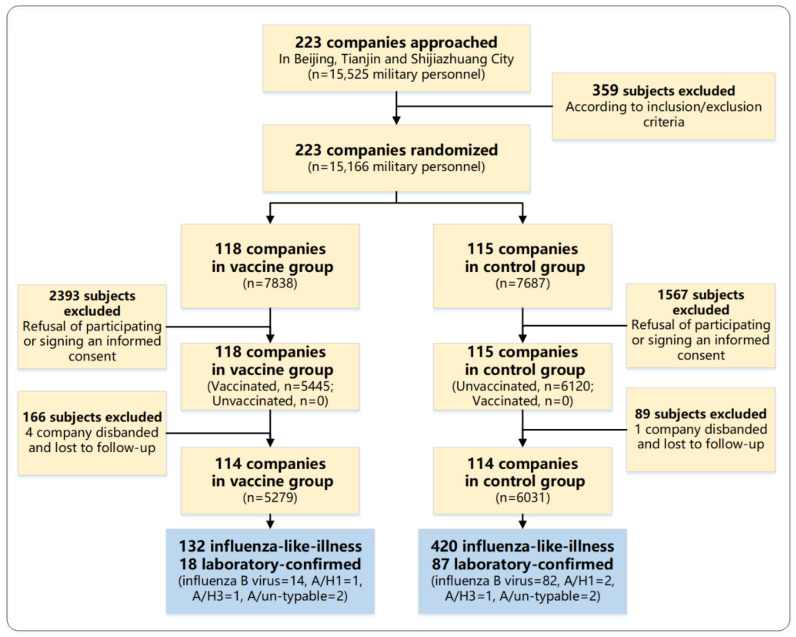
Participant disposition.

**Figure 2 vaccines-11-01439-f002:**
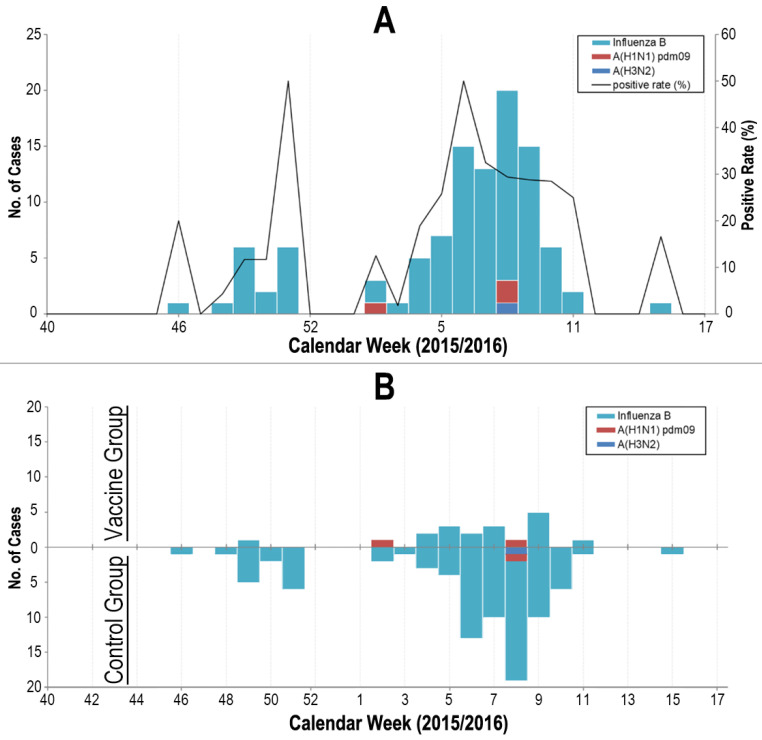
Occurrence of influenza A and B in the study cohort from 1 October 2015 to 30 April 2016. Panel (**A**). Occurrence of influenza in the whole study cohort. Panel (**B**). Occurrence of influenza by vaccine group (upper quadrant of x-axis) and control group (lower quadrant of x-axis).

**Figure 3 vaccines-11-01439-f003:**
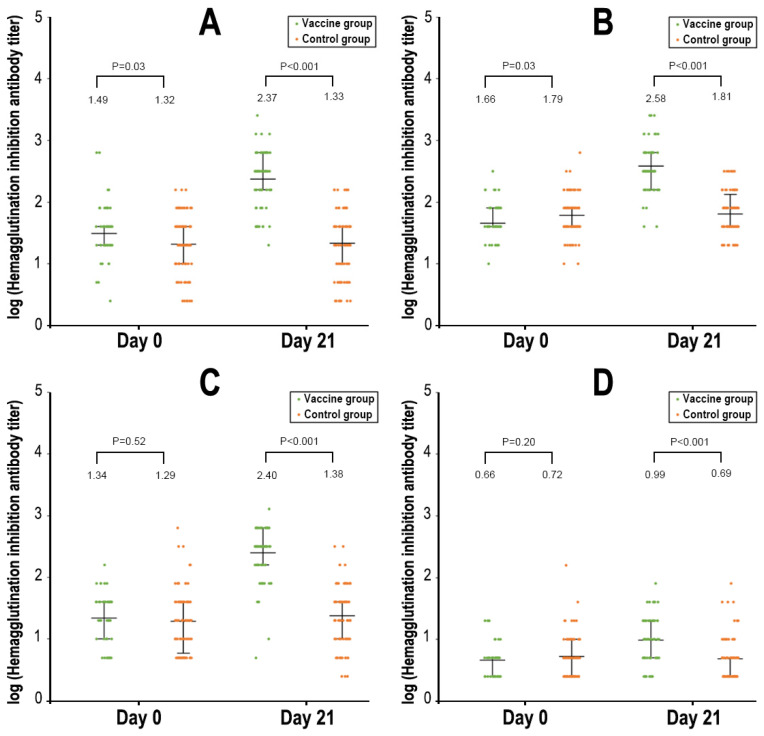
Hemagglutination inhibition antibody titers to different strains of influenza virus. Panel (**A**). A/H1N1. Panel (**B**). A/H3N2. Panel (**C**). B/Yamagata. Panel (**D**). B/Victoria.

**Table 1 vaccines-11-01439-t001:** Baseline characteristics of study participants and cluster characteristics.

Characteristics	Influenza Vaccine Group 114 Clusters (n = 5279)	Control Group 114 Clusters (n = 6031)
Individual level		
Age, median (IQR), yr	21 (19–24)	21 (19–24)
Age groups, y, No. (%)		
18–22	3600 (68.2)	3966 (65.8)
23–30	1506 (28.5)	1844 (30.6)
31–38	173 (3.3)	221 (3.7)
Female sex, No. (%)	43 (0.8)	22 (0.4)
Ethnicity, No. (%)		
Han	5069 (96.0)	5795 (96.1)
Other	210 (4.0)	236 (3.9)
Educational attainment, No. (%)		
Primary school or illiterate	748 (14.2)	738 (12.2)
High school	3236 (61.3)	3785 (62.8)
College level	852 (16.1)	1030 (17.1)
Undergraduate and above	443 (8.4)	478 (7.9)
Cluster level, median (IQR)		
All persons—no./cluster	60 (43–87)	63 (36–89)
Enrolled participants—No./cluster	36 (20–65)	43 (27–83)
Vaccinated subjects—No./cluster	36 (20–65)	0 (0–0)

**Table 2 vaccines-11-01439-t002:** Occurrence of influenza-like illness (ILI), laboratory-confirmed influenza, and the protective effectiveness for vaccinated subjects of influenza-vaccine-immunized clusters.

Variable	Vaccine Group (n = 5279)		Control Group (n = 6031)		Protective Effectiveness of Influenza Vaccination (95% CI)
	No. of Subjects	Incidence, %	No. of Subjects	Incidence, %	
Influenza-like illness	132	2.50 (2.10–2.96)	420	6.96 (6.33–7.64)	64.1 (56.2 to 70.6)
Laboratory-confirmed influenza	18	0.34 (0.20–0.54)	87	1.44 (1.16–1.78)	76.4 (60.7 to 85.8)
Influenza B virus	14	0.27 (0.15–0.44)	82	1.36 (1.08–1.68)	80.5 (65.6 to 88.9)
Influenza A virus	4	0.08 (0.02–0.19)	5	0.08 (0.03–0.19)	8.6 (−241 to 75.5)
Influenza A subtype H1	1	0.02 (0.00–0.11)	2	0.03 (0.00–0.12)	42.9 (−530 to 94.8)
Influenza A subtype H3	1	0.02 (0.00–0.11)	1	0.02 (0.00–0.09)	−14.2 (−1727 to 92.9)
Influenza A un-typable	2	0.04 (0.00–0.14)	2	0.03 (0.00–0.12)	−14.2 (−711 to 83.9)

## Data Availability

De-identified individual participant-level data will be available upon written request to the corresponding author following publication.

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
