# Peer review of "Effect of Influenza Vaccination on Rate of Influenza Virus Infection in Chinese Military Personnel, 2015–2016: A Cluster Randomized Trial"

_vaccines, 2023, doi:10.3390/vaccines11091439_

Round 1

Reviewer 1 Report

The issue is of high public health importance over the world.  

There are quite few well documented study related to the use of the anti influenza inactivated vaccines among military population groups .

So far, the level of effectiveness of existing vaccines is not quite high and most countries are quite hesitant in their vaccine regulations.  

I suggest the authors to give more attention to WHO recommandations:

- in the introduction ( pinpoint the challenges for vaccine strategies formulation especially in the confined settings of military populations, any information on the morbidity /mortality impact  due to influenza in military poulations over the world and cost benefits opportunities for including  anti influenza vaccines into the vaccine agenda?...)

- in the discussions ( any references related to the differences of effectivenes rates and consideration on reasons of differences/ Survey methodology?Type of vaccine? delivery conditions ?...

- in the conclusion: emphasize the criteria for decision making  in vaccination programmes in  military population   

Conclusion.

Relevant public health issue, good study design and methodology, results well presented, fair discussions on the key findings.

For the reader, it would be interesting to know more on the decision making challenges, at the ligfht of WHO experts group recommandations.

This could be given in adding some information points and consideration in the ingtroduction, the discussion section and the conclusions.

Author Response

We appreciate the reviewer’s comments. We added descriptions of challenges in military populations in the introduction (Please refer to pages 1-2, lines 38-55), discussed factors influencing vaccine effectiveness in the discussion (Please refer to page 11, lines 328-331), and added some decision-making suggestions for military population in the conclusion (Please refer to page 13, lines 423-425), accordingly. Thank you.

Reviewer 2 Report

A cluster randomized trial to reveal the protective effect of trivalent inactivated influenza vaccine (TIV) was conducted by Li et al. The study reported the TIV is moderately effective, highly immunogenic, and generally safe to use in healthy young adults. Overall, the study was designed and presented properly. The data of post-licensure effectiveness of TIV in the active-duty servicemen is helpful for the vaccine implementation. The design of cluster randomization should be a merit of the study; however, the indirect effect and overall effect could not be evaluated due to the dropout, which is a pity. Several issues remain to be clarified in the manuscript:

1.     In this study, the number of ILI reported in the vaccine group and control group varies greatly. The protective effect against ILI was showed, and it could be inferred that the protective effect against non-influenza ILI was about 61%, however, which is doubted. One concern is that the bias in ILI surveillance exist, which might be caused by the design of open label. Are there any actions conducted to control the bias in the study? Do the authors have more evidence showed the protective effect of influenza vaccine against non-influenza ILI? I suggest clarifying the possible bias in the open label trial in the discussion.

2.     In the sample size calculation, please clarify the coefficient of variation of cluster sizes.

3.     For those breakthrough cases, the analysis on symptom was suggested to perform for understanding the effect at alleviating symptoms or shortening the disease course.

Author Response

Q1. In this study, the number of ILI reported in the vaccine group and control group varies greatly. The protective effect against ILI was showed, and it could be inferred that the protective effect against non-influenza ILI was about 61%, however, which is doubted. One concern is that the bias in ILI surveillance exist, which might be caused by the design of open label. Are there any actions conducted to control the bias in the study? Do the authors have more evidence showed the protective effect of influenza vaccine against non-influenza ILI? I suggest clarifying the possible bias in the open label trial in the discussion.

Response: Accepted. We added a limitation and discussed potential biases caused by the open-label trial design in the revised manuscript (Please refer to pages 12-13, lines 410-414). Thank you.

Q2. In the sample size calculation, please clarify the coefficient of variation of cluster sizes.

Response: We appreciate the reviewer’s comments. In the sample size calculation, we assumed a coefficient of variation of cluster sizes to be 0.832 based on our past experience. With the prior parameter assumptions and following formulation, we calculated that a sample size of 97 clusters would be needed for each group.

SScluster RCT=SSstandard RCT×DEunequal

SSstandard RCT=(Z1-2/α+Z1-β)2[p1(1-p1)+p2(1-p2)]/(p1-p2)2

DEunequal=1+[(1+cv2)m-1]ρ

cv = coefficient of variation of cluster size

m = mean cluster size

ρ= intra-cluster correlation coefficient

Allowing for a dropout rate of 10%, a final sample size of 108 clusters (n=7,492) would be needed for each group. We have made this point clear in the revised manuscript (Please refer to page 5, lines 205-206). Thank you.

Q3. For those breakthrough cases, the analysis on symptom was suggested to perform for understanding the effect at alleviating symptoms or shortening the disease course.

Response: Accepted. We compared the clinical signs/symptoms & the disease course of the breakthrough cases in the vaccine group with that of the 87 laboratory-confirmed influenza in the control group and added one supplementary table 2 in the revised manuscript, accordingly. Our analyses showed that participants who received one doses of TIV experienced less frequent headache (5.6% vs. 37.9%, p=0.006) and myalgia (0% vs. 40.2%, p=0.001) when they had laboratory-confirmed influenza during the follow-up periods. (Please refer to page 8, lines 272-275 & Supplementary Table 2). Thank you.

Reviewer 3 Report

Very high-quality manuscript profiling the effectiveness of trivalent influenza vaccine (TIV) in military personnel during the 2015 – 2016 flu season in China.

No concerns about the methodology, results, or discussion presented. These results are very interesting and merit publication. However, for completeness, authors need to also include a comment on the current epidemiology of B/Yamagata. There have been no detections of that strain since 2020 and it may be extinct.

Authors correctly identify TIV vaccine as effective, immunogenic, and safe, but must address/discuss that >90% of incidence in the trial is associated with a strain that may no longer be in circulation. If B/Yamagata is no longer in circulation and is (potentially) dropped from the flu vaccine, how would that affect this study's conclusions?

Author Response

Q1. No concerns about the methodology, results, or discussion presented. These results are very interesting and merit publication. However, for completeness, authors need to also include a comment on the current epidemiology of B/Yamagata. There have been no detections of that strain since 2020 and it may be extinct.

Response: We appreciate the reviewer’s comments. We added some descriptions on the current epidemiology of B/Yamagata in the revised manuscript. (Please refer to page 12, lines 372-378). 

Q2. Authors correctly identify TIV vaccine as effective, immunogenic, and safe, but must address/discuss that >90% of incidence in the trial is associated with a strain that may no longer be in circulation. If B/Yamagata is no longer in circulation and is (potentially) dropped from the flu vaccine, how would that affect this study's conclusions?

Response: We appreciate the reviewer’s comments. The WHO recommend the antigenic composition of TIV twice a year. The component of Anflu® is updated every year from influenza viruses recommended by WHO for northern hemisphere. If B/Yamagata is no longer in circulation, it would have favorable implications for annual influenza vaccine development of Anflu® as some quadrivalent influenza vaccine (QIV) might need to return to TIV with two IAV strains and a B/Victoria strain in the formulation.